# Coordination Polymers Bearing Angular 4,4′-Oxybis[*N*-(pyridin-3-ylmethyl)benzamide] and Isomeric Dicarboxylate Ligands: Synthesis, Structures and Properties

**DOI:** 10.3390/molecules30153283

**Published:** 2025-08-05

**Authors:** Yung-Hao Huang, Yi-Ju Hsieh, Yen-Hsin Chen, Shih-Miao Liu, Jhy-Der Chen

**Affiliations:** 1Department of Chemistry, Chung Yuan Christian University, Chung Li, Taoyuan City 320314, Taiwan; a0987198719@gmail.com (Y.-H.H.); yovr2840@gmail.com (Y.-J.H.); sunny911228@gmail.com (Y.-H.C.); 2Center for General Education, Hsin Sheng Junior College of Medical Care and Management, Longtan 32544, Taiwan

**Keywords:** coordination polymer, X-ray structure, topology, angular ligand

## Abstract

Reactions of the angular 4,4′-oxybis[*N*-(pyridin-3-ylmethyl)benzamide] (**L**) with dicarboxylic acids and transition metal salts afforded non-entangled {[Cd(**L**)(1,3-BDC)(H_2_O)]∙2H_2_O}_n_ (1,3-BDC = 1,3-benzenedicarboxylic acid), **1**; {[Cd(**L**)(1,4-HBDC)(1,4-BDC)_0.5_]∙2H_2_O}_n_ (1,4-BDC = 1,4-benzenedicarboxylic acid), **2**; {[Cu_2_(**L**)_2_(1,3-BDC)_2_]∙1.5H_2_O}_n_, **3**; {[Ni(**L**)(1,3-BDC)(H_2_O)]∙2H_2_O}_n_, **4**; {[Zn(**L**)(1,3-BDC)]∙4H_2_O}_n_, **5**; {[Zn(**L**)(1,4-BDC)]∙2H_2_O}_n_, **6**; and [Cd_3_(**L**)_2_(1,4-BDC)_3_]_n_, **7**, which have been structurally characterized by using single-crystal X-ray diffraction. Complexes **1**–**5** and **7** are 2D layers, giving (6^4^·8·10)(6)-2,4L3, (4^2^·8^2^·10^2^)(4^2^·8^4^)_2_(4)_2_, (4·5·6)(4·5^5^·6^3^·7)-3,5L66, (6^4^·8·10)(6)-2,4L3, interdigitated (8^4^·12^2^)(8)_2_-2,4L2 and (3^6^·4^6^·5^3^)-**hxl** topologies, respectively, and **6** is a 1D chain with the (4^3^·6^2^·8)(4)-2,4C3 topology. The factors that govern the structures of **1**–**7** are discussed and the thermal properties of **1**–**7** and the luminescent properties of complexes **1**, **2**, **5** and **6** are investigated. The stabilities of complexes **1** and **5** toward the detection of Fe^3+^ ions are also evaluated.

## 1. Introduction

The research on coordination polymers (CPs) has been rapidly growing in recent years because of the formation of interesting topological structures and the wide range of potential applications, predominantly in catalysis, gas storage and separation, ion exchange, magnetism, luminescence, sensors and so on [1,2,3,4,5]. It is well known that the structural types of the resulting CPs are influenced by various factors, including the identities of the organic linker, metal ion and counter ion, as well as the reaction conditions such as temperature, metal to ligand ratio and solvent system. Therefore, controlling the appropriate factors to construct CPs with desired structural types and properties has become an exciting topic in the crystal engineering of CPs [6,7,8,9,10,11,12]. The structures of many of the CPs are quite fascinating because of the formation of entanglement, resulting from the presence of large free voids in a single network. However, entanglement of CPs involving interpenetration, polycatenation and self-catenation may limit the pore sizes of CPs that are important in achieving high gas storage and separation [2].

The flexible bis-pyridyl-bis-amide (bpba) ligands have the advantages of adopting different ligand conformations and tending to form entangled structures to maximize the packing efficiencies of the crystal structures consequently prepared. In combination with angular dicarboxylic acids, entangled CPs containing *N*,*N′*-di(4-pyridyl)suberoamide have shown that the angular dicarboxylate ligands display significant effect on the degree of interpenetration. On the other hand, the use of angular dicarboxylate ligands and flexible *N,N′*-di(4-pyridyl)sebacoamide or *N,N′*-di(4-pyridyl)adipoamide afforded five entangled CPs and one non-entangled one, whereas using the rigid *N,N′*-bis(4-pyridylmethyl)bicyclo(2,2,2,)oct-7-ene-2,3,5,6-tetracarboxylic diamide gave three non-entangled and one entangled CP [10]. We thus proposed that Co(II) CPs containing flexible bpba and angular dicarboxylate ligands are more likely to form entangled CPs than those with rigid ones [10]. Figure 1 shows the structures of the corresponding organic ligands.

With the above-mentioned background information, to investigate the structure-directing role of the angular bpba in forming the entangled CPs with mixed ligands, we have prepared 4,4′-oxybis(*N*-(pyridin-3-ylmethyl)benzamide (**L**) to explore its coordination chemistry with metal ions and dicarboxylic acids. Herein, the synthesis and structures of {[Cd(**L**)(1,3-BDC)(H_2_O)]∙2H_2_O}_n_ (1,3-BDC = 1,3-benzenedicarboxylic acid), **1**; {[Cd(**L**) (1,4-HBDC)(1,4-BDC)_0.5_]∙2H_2_O}_n_ (1,4-BDC = 1,4-benzenedicarboxylic acid), **2**; {[Cu_2_(**L**)_2_(1,3-BDC)_2_]∙1.5H_2_O}_n_, **3**; {[Ni(**L**)(1,3-BDC)(H_2_O)]∙2H_2_O}_n_, **4**; {[Zn(**L**)(1,3-BDC)]∙4H_2_O}_n_, **5**; {[Zn(**L**)(1,4-BDC)]∙2H_2_O}_n_, **6;** and [Cd_3_(**L**)_2_(1,4-BDC)_3_]_n_, **7** form the subject of this report. The luminescent properties of complexes **1**, **2**, **5** and **6** are reported, and the stabilities of **1** and **5** toward the detection of the Fe^3+^ ions are evaluated.

## 2. Results and Discussion

### 2.1. Structure of {[Cd(**L**)(1,3,-BDC)(H_2_O)]∙2H_2_O}_n_, ***1***

A single-crystal X-ray diffraction analysis shows that **1** crystallizes in the triclinic space group *P*ī. The asymmetric unit consists of one Cd(II) cation, one **L** ligand, one 1,3-BDC^2−^ ligand, one coordinated water molecule and two cocrystallized water molecules. Figure 2a depicts the coordination environment of **1**, which is six-coordinated by three oxygen atoms from two 1,3-BDC^2−^ ligands [Cd-O = 2.2733(13)–2.4849(12) Å], one oxygen atom from the coordinated water molecule [Cd-O = 2.3393(14) Å] and two pyridyl nitrogen atoms from two **L** ligands [Cd-N = 2.2697(14)–2.3427(15) Å], resulting in a distorted octahedral geometry. The Cd(II) ions are linked by 1,3-BDC^2−^ and **L** ligands to form a 2D layer. If the Cd(II) atoms are considered as four-connected nodes, the **L** ligands as two-connected nodes and the 1,3-BDC^2−^ ligands as linkers, the structure of **1** can be simplified as a 2,4-connected 2D net with the (6^4^·8·10)(6)-2,4L3 topology, Figure 2b, determined by using ToposPro [13]. The 2D layers of **1** are supported by the N-H---O hydrogen bonds from the amide groups to the carboxylate oxygen atoms [H---O = 2.026 Å; ∠N-H---O = 160.98°] and cocrystallized water molecules [H---O = 2.136 Å; ∠N-H---O = 152.51°]. O-H---O hydrogen bonds from the cocrystallized water molecules to the carboxylate oxygen atoms [H---O = 1.945 and 2.382 Å, ∠O-H---O = 164.06 and 155.62°] and amide oxygen atoms [H---O = 1.945 and 2.056 Å, ∠O-H---O = 170.52 and 152.09°], as well as from the coordinated water molecules to the carboxylate groups [H---O = 1.988 Å, ∠O-H---O = 170.22°] and cocrystallized water molecules [H---O = 2.111 Å, ∠O-H---O = 152.83°] are also observed.

### 2.2. Structure of {[Cd(**L**)(1,4-HBDC)(1,4-BDC)_0.5_]∙2H_2_O}_n_, ***2***

Complex **2** crystallizes in the triclinic space group *P*ī and the asymmetric unit consists of one Cd(II) cation, one **L** ligand, one 1,4-HBDC^−^ ligand, half of an 1,4-BDC^2−^ ligand and two cocrystallized water molecules. Figure 3a demonstrates the coordination environment of **2**, which is seven-coordinated by five oxygen atoms from three 1,4-BDC^2−^ ligands [Cd-O = 2.297(4)–2.604(4) Å] and two pyridyl nitrogen atoms from two **L** ligands [Cd-N = 2.277(5)–2.286(6) Å], resulting in a distorted pentagonal bipyramidal geometry. The Cd(II) ions are linked by 1,3-BDC^2−^ and **L** ligands to form a 2D layer. If the Cd(II) atoms and the 1,4-BDC^2−^ ligands are considered as four-connected nodes, the **L** ligands as two-connected nodes and the 1,4-HBDC^−^ ligands serve as linkers, the structure of **2** can be regarded as a 2,4,4-connected 2D net with the (4^2^·8^2^·10^2^)(4^2^·8^4^)_2_(4)_2_ topology, Figure 3b. Moreover, if dinuclear Cd(II) units are considered as four-connected nodes, while the organic ligands as linkers, the structure of **2** can be further simplified as 2D net with (4^4^∙6^2^)-**sql** topology, Figure 3c. The 2D structures of **2** are linked by the N-H---O hydrogen bonds from the amide groups to the adjacent amide oxygen atoms [H---O = 2.174 and 2.187 Å; ∠N-H---O = 156.76 and 135.69°]. O-H---O hydrogen bonds from the cocrystallized water molecules to the amide oxygen atoms [H---O = 2.375 and 2.173 Å, ∠O-H---O = 128.60 and 137.83°] and cocrystallized water molecules [H---O = 1.821 Å, ∠O-H---O = 133.76°], as well as from the carboxylate hydrogen atom to the carboxylate groups [H---O = 1.823Å, ∠O-H---O = 157.45°] are also observed.

### 2.3. Structure of {[Cu_2_(**L**)_2_(1,3-BDC)_2_]∙1.5H_2_O}_n_, ***3***

Crystals of complex **3** conform to the triclinic space group Pī. The asymmetric unit consists of two Cu(II) cations, two **L** ligands, two 1,3-BDC^2−^ ligands, and one and a half cocrystallized water molecules. Figure 4a depicts the coordination environments of the Cu(II) cations, and both of Cu(1) and Cu(2) are five-coordinated. While Cu(1) is coordinated by three oxygen atoms from three 1,3-BDC^2−^ ligands [Cu(1)-O = 1.9504(17)–2.3684(19) Å] and two pyridyl nitrogen atoms from two **L** ligands [Cu(1)-N = 2.016(2)–2.023(2) Å], Cu(2) is coordinated by three oxygen atoms from three 1,4-BDC^2−^ ligands [Cu(2)-O = 1.9368(16)–2.396(2) Å] and two pyridyl nitrogen atoms from two **L** ligands [Cu(2)-N = 2.011(2)–2.026(2) Å], resulting in distorted square pyramidal geometries with τ_5_ values of 0.29 and 0.35, respectively [14]. The Cu(II) ions are linked together by 1,3-BDC^2−^ to form a 2D layer. If the Cu(II) ions are considered as five-connected nodes, the 1,3-BDC^2−^ ligands as three-connected nodes and the **L** ligands as two-connected nodes, the structure of **3** can be simplified as a 2,3,5-connected 2D net with the (4·6·8)(4·6^5^·8^4^)(6) topology, Figure 4b. Moreover, if the **L** ligands are considered as linkers, the structure of **3** can be regarded as a 3,5-connected 2D net with the (4·5·6)(4·5^5^·6^3^·7)-3,5L66 topology, Figure 4c. The 2D layers of **3** are linked by the N-H---O hydrogen bonds from the amide groups to the adjacent amide oxygen atoms [H---O = 2.197 Å; ∠N-H---O = 136.47°] and cocrystallized water molecules [H---O = 1.915 Å; ∠N-H---O = 175.23°]. O-H---O hydrogen bonds from the cocrystallized water molecules to the amide oxygen atoms [H---O = 1.843 Å, ∠O-H---O = 129.51°] are also observed.

### 2.4. Structure of {[Ni(**L**)(1,3-BDC)(H_2_O)]∙2H_2_O}_n_, ***4***

The structure of complex **4** was solved in the triclinic space group *P*ī. The asymmetric unit consists of one Ni(II) cation, one **L** ligand, one 1,3-BDC^2−^ ligand, one coordinated water molecule and two cocrystallized water molecules. Figure 5a gives the coordination environment of **4**, which is six-coordinated by four oxygen atoms from three 1,3-BDC^2−^ ligands [Ni-O = 2.0344(10)–2.1563(10) Å] and two pyridyl nitrogen atoms from two **L** ligands [Ni-N = 2.0658(12)–2.0895(13) Å], resulting in a distorted octahedral geometry. The Ni(II) cation ions are linked by 1,3-BDC^2−^ and **L** ligands to form a 2D layer. If the Ni(II) atoms are considered as four-connected nodes, the **L** ligands as two-connected nodes and the 1,3-BDC^2−^ ligands as linkers, the structure of **4** can be regarded as a 2,4-connected 2D net with the (6^4^·8·10)(6)-2,4L3 topology, Figure 5b. Moreover, if the **L** ligands are also considered as linkers, the structure of **4** can be further simplified as a 4-connected 2D net with a (4^4^)-**sql** topology. The 2D layers of **4** are supported by the N-H---O hydrogen bonds from the amide groups to the carboxylate oxygen atoms [H---O = 1.978 Å; ∠N-H---O = 165.49°] and cocrystallized water molecules [H---O = 2.287 Å; ∠N-H---O = 145.62°]. O-H---O hydrogen bonds from the cocrystallized water molecules to the carboxylate oxygen atoms [H---O = 2.019 Å, ∠O-H---O = 159.26°] and amide oxygen atoms [H---O = 2.130, 2.194 and 1.930 Å, ∠O-H---O = 158.11, 157.67 and 174.82°], as well as from the coordinated water molecules to the carboxylate groups [H---O = 1.952 Å, ∠O-H---O = 158.29°] and cocrystallized water molecules [H---O = 2.088 Å, ∠O-H---O = 138.01°] are also observed.

### 2.5. Structure of {[Zn(**L**)(1,3-BDC)]∙4H_2_O}_n_, ***5***

Complex **5** crystallizes in the monoclinic space group P2_1_/n, with the asymmetric unit comprising one Zn(II) cation, one **L** ligand, one 1,3-BDC^2−^ ligand and four cocrystallized water molecules. Figure 6a shows the coordination environment of **5**, which is four-coordinated by two oxygen atoms from two 1,3-BDC^2−^ ligands [Zn-O = 1.9293(16)–1.9419(14) Å] and two pyridyl nitrogen atoms from two **L** ligands [Zn-N = 2.043(2)–2.111(2) Å], resulting in a distorted tetrahedral geometry. The Zn(II) cation ions are linked by 1,3-BDC^2−^ and **L** ligands to form a 2D layer. If the Zn(II) ions are considered as four-connected nodes and the 1,3-BDC^2−^ and **L** ligands as two-connected nodes, the structure of **5** can be simplified as a 2,4-connected 2D net with the (8^4^·12^2^)(8)_2_-2,4L2 topology, Figure 6b, showing pairs of interdigitated 2D layers, Figure 6c,d. The interdigitation is presumably due to the angular shapes of the organic ligands that maximize the packing efficiency. The 2D layers of **5** are supported by the N-H---O hydrogen bonds from the amide groups to the carboxylate oxygen atoms [H---O = 2.222 Å; ∠N-H---O = 139.46°] and cocrystallized water molecules [H---O = 1.959 Å; ∠N-H---O = 160.02°]. O-H---O hydrogen bonds from the cocrystallized water molecules to the carboxylate oxygen atoms [H---O = 2.058 Å, ∠O-H---O = 158.03°], amide oxygen atoms [H---O = 2.049 and 1.941 Å, ∠O-H---O = 153.69 and 168.23°] and cocrystallized water molecules [H---O = 2.337, 2.353, 1.918, 1.962 and 2.407 Å, ∠O-H---O = 158.27, 121.16, 165.76, 174.40 and 140.26°] are also observed.

### 2.6. Structure of {[Zn(**L**)(1,4-BDC)]∙2H_2_O}_n_, ***6***

Crystals of complex **6** conform to the monoclinic space group *P*2_1_/*c*. The asymmetric unit consists of one Zn(II) cation, one **L** ligand, one 1,4-BDC^2−^ ligand and two cocrystallized water molecules. Figure 7a depicts the coordination environment of **6**, which is five-coordinated by three oxygen atoms from two 1,4-BDC^2−^ ligands [Zn-O = 1.939(2)–2.421(4) Å] and two pyridyl nitrogen atoms from two **L** ligands [Zn-N = 2.033(3)–2.063(3) Å], resulting in a distorted square pyramidal geometry (τ_5_ = 0.12). The Zn(II) cation ions are linked by 1,3-BDC^2−^ and **L** ligands to form a 1D chain. If the Zn(II) ions are considered as four-connected nodes, the **L** ligands as two-connected nodes and the 1,3-BDC^2−^ ligands as linkers, the structure of **6** can be further simplified as a 2,4-connected 1D net with the (4^3^·6^2^·8)(4)-2,4C3 topology, Figure 7b. The 1D chains of **6** are supported by the N-H---O hydrogen bonds from the amide groups to cocrystallized water molecules [H---O = 2.020 and 2.012 Å; ∠N-H---O = 149.23 and 156.39°]. O-H---O hydrogen bonds from the cocrystallized water molecules to the carboxylate oxygen atoms [H---O = 1.928 and 2.137 Å, ∠O-H---O = 170.01 and 166.53°] and amide oxygen atoms [H---O = 1.906 and 1.861 Å, ∠O-H---O = 165.43 and 171.71°] are also observed.

### 2.7. Structure of [Cd_3_(**L**)_2_(1,4-BDC)_3_]_n_, ***7***

Complex **7** crystallizes in the monoclinic space group *C*2/*c*. The asymmetric unit consists of one and a half of a Cd(II) cation, one **L** ligand and one and a half of a 1,4-BDC^2−^ ligand. Figure 8a demonstrates the coordination environments about the Cd(II) ions of **7**. Both Cd(1) and Cd(2) are six-coordinated, resulting in distorted octahedral geometries. While the Cd(1) is coordinated by two pyridyl nitrogen atoms from two **L** ligands [Cd-N = 2.378(4)–2.382(4) Å] and four oxygen atoms from three 1,4-BDC^2−^ ligands [Cd-O = 2.247(4)–2.464(3) Å], the Cd(2) is coordinated by six oxygen atoms from six 1,4-BDC^2−^ ligands [Cd-O = 2.166(3)–2.402(3) Å]. The Cd(II) ions are chelated and bridged by the 1,4-BDC^2−^ ligands to form linear trinuclear unites, which are also linked by the **L** ligands to form 2D layers. If Cd(1) and Cd(2) are considered as five- and six-connected nodes, the two independent 1,4-BDC^2−^ ligands as four-connected nodes and the **L** ligands as linkers, the structure of **7** can be regarded as a 4,4,5,6-connected 2D net with the (4^2^·6·7^3^)(4^3^·5^2^·6^4^·7)_2_(4^3^·5^3^)_2_(4^8^·6^6^·7) topology (standard representation), Figure 8b. Moreover, if trinuclear Cd(II) units are considered as six-connected nodes and the organic ligands as linkers, the structure of **7** can be further simplified as a 6-connected 2D net with the (3^6^·4^6^·5^3^)-**hxl** topology (cluster representation), Figure 8c. The 2D layers of **7** are supported by the N-H---O hydrogen bonds from the amide groups to the carboxylate oxygen atoms [H---O = 1.905 Å; ∠N-H---O = 152.72°] and amide oxygen atoms [H---O = 2.097 Å; ∠N-H---O = 176.81°].

### 2.8. Ligand Conformations and Coordination Modes

The relative orientations of the C=O groups and pyridyl nitrogen atoms can be evaluated. When the C=O group and pyridyl nitrogen atom point to the same orientation, it is defined as *syn*, and if the relative orientation exhibits the opposite orientation, it is defined as *anti*. Accordingly, the conformations of the **L** ligands are listed in Table 1 as well as the coordination mode of the polycarboxylate ligands. While all of the **L** ligands in complexes **1**–**7** bridge two metal ions through two pyridyl nitrogen atoms, resulting in various ligand conformations, the dicarboxylate ligands chelate and/or bridge with two to four metal ions with diverse coordination modes.

### 2.9. Powder X-Ray Analysis

The phase purities of complexes **1**–**7** were evaluated by measuring their powder X-ray diffraction (PXRD) patterns. As shown in Appendix A, the peak positions of the experimental PXRD patterns of **1**–**7** match well with their corresponding simulated ones, suggesting the good bulk purities of these CPs.

### 2.10. Thermal Properties

The thermal gravimetric analysis (TGA) was carried out to investigate the thermal decompositions of complexes **1**–**7**. The samples were heated from 30 to 800 °C at a rate of 10 °C min^−1^ under a N_2_ atmosphere, Appendix A. Table 2 lists the thermal parameters of complexes **1**–**7**, revealing that **1**–**6** suffer two-step decomposition involving loss of solvent and loss of organic ligands, while only loss of organic ligands is observed for **7**.

### 2.11. Luminescent Properties

The metal centers of the CPs have the ability to shift, enhance and quench the emission wavelength of organic ligands through coordination, which can be regarded as potential materials for luminescent applications. The luminescent properties of complexes **1**, **2**, **5**, **6** and **7** and free ligands **L**, 1,3-H_2_BDC and 1,4-H_2_BDC were thus investigated at room temperature in the solid state. Table 3 summarizes the maximum excitation and emission wavelengths of **1**, **2**, **5**, **6**, **7**, **L**, 1,3-H_2_BDC and 1,4-H_2_BDC. The **L**, 1,3-H_2_BDC and 1,4-H_2_BDC show emissions at 398, 355 and 385 nm, respectively, Appendix A, which can be attributed to the intraligand (IL) n → π* or π → π* transitions. Due to the d^10^ electronic configuration in Cd(II) and Zn(II), they hardly undergo oxidation or reduction [15,16]. The emissions of **1**, **2**, **5**, **6** and **7** are thus not due to ligand-to-metal charge transfer (LMCT) or metal-to-ligand charge transfer (MLCT). Therefore, these emissions may be ascribed to intraligand or ligand-to-ligand charge transfer (LLCT). The blue and/or red shifts of **1**, **2**, **5**, **6** and **7** with respect to their corresponding organic ligands may be due to the different ligand conformations and coordination modes that result in different structural types.

### 2.12. Metal Ion Detection

Complexes **1** and **5** are two diverse CPs with the same organic ligands but different metal ions, Cd(II) and Zn(II), which provide a unique opportunity to investigate the role of metal identity in determining the structural stabilities of thus CPs toward the detection of Fe^3+^ ions. To compare the structural roles of **1** and **5** toward the detection of the metal ions, 50 mg samples of each of the complexes were added into the 10 mL of the aqueous solutions of Fe(NO_3_)_3_ or M(OAc)_2_ with a concentration of 1 × 10^−2^ M. After 12 h, the complexes were filtered and washed with water, and the solid-state emission spectra were then measured at room temperature, Appendix A. Figure 9 depicts the average luminescent intensities of complexes **1** and **5** upon immersion in the solutions of various metal ions, revealing significant quenching effect in the case of Fe^3+^ ions, and about 94.8 and 98.4% of the luminescent intensities of complexes **1** and **5** are quenched.

### 2.13. Luminescent Quenching Mechanism

Luminescence quenching can be possibly associated with framework collapse, cation exchange and competitive absorption of the excited energies between the incoming metal ion and the host complex [17,18,19,20,21,22,23,24]. To evaluate the probable mechanism of the luminescence quenching by the metal ion, the PXRD patterns of **1** and **5** were measured after immersing in the 1 × 10^−2^ M aqueous solutions of various metal ions for 12 h. As shown in Appendix A, the crystallinity of complex **1** before and after experiment remained intact for three rounds, suggesting that the structural integrity of **1** is not compromised in the presence of the analyte ions. On the other hand, the crystallinity of complex **5** collapsed upon immersion in the solutions of Fe^3+^ ions, Figure 10, indicating the quenching mechanism can be mainly attributed to the disintegration of the framework. The different metal identities in **1** and **5** that result in 2D layers with the 2,4L3 and interdigitated 2,4L2 topologies, respectively, thus play different roles in determining their chemical stabilities toward the sensing of the Fe^3+^ ions. The interdigitation of the 2D layer of complex **5** may be one of the factors that leads to the decomposition of **5** upon the attack of the Fe^3+^ ions.

The UV–vis absorption spectrum of Fe^3+^ in aqueous solution and the corresponding excitation and emission spectra of complex **1** are shown in Figure 11. It is seen that the absorption spectrum of Fe^3+^ ions is partially overlapped with the excitation spectra of **1**, demonstrating that the excitation energy of **1** can be absorbed by the Fe^3+^ ions. Therefore, the luminescence quenching of **1** by the Fe^3+^ ions is most probably due to the competitive energy absorption [17].

The quenching effect of Fe^3+^ ion on the emission of complex **1** can be validated by using the Stern–Volmer equation: I_0_/I = 1 + K_sv_ × [Q]. I_0_ and I are the emission intensities of 1 without and with Fe^3+^ ion, respectively; [Q] is the concentration of Fe^3+^ ion; and K_sv_ is the quenching constant [18]. As shown in Figure 12 and Appendix A, the titration curve for Fe^3+^ ions in **1** is basically linear at low concentrations, affording a linear correlation coefficient (R^2^) of 0.9995 and a Stern−Volmer constant (K_sv_) value of 1.14 × 10^4^ M^−1^, respectively. Moreover, the detection limit 3*σ*/*k* gives a value of 1.06 × 10^−4^ M, where *σ* is the standard deviation from the blank measurements and *k* is the absolute value of the calibration curve at lower concentrations [19]. For a comparison, the K_sv_ value of the **L**-based Cd(II) CP toward the detection of Fe^3+^ ion is 1.9 × 10^4^ M^−1^ for {[Cd_2_(L)(1,4-NDC)_2_]·EtOH}_n_ (1,4-H_2_NDC = naphthalene-1,4-dicarboxylic acid), indicating that the identity of the dicarboxylate ligand that results in different structural types may affect the sensing effect of the **L**-based CPs toward the detection of Fe^3+^ ion. Moreover, the recyclability test reveals that complex **1** can be reused most probably for five regeneration cycles, on the basis of the PXRD patterns measured after each test, Appendix A.

## 3. Experimental Section

### 3.1. General Procedures

Elemental analyses of (C, H, N) were performed on a PE 2400 series II CHNS/O (PerkinElmer Instruments, Shelton, CT, USA) or an Elementar Vario EL-III analyzer (Elementar Analysensysteme GmbH, Hanau, Germany). Infrared spectra were obtained from a JASCO FT/IR-460 plus spectrometer with pressed KBr pellets (JASCO, Easton, MD, USA). Powder X-ray diffraction patterns were carried out with a Bruker D8-Focus Bragg–Brentano X-ray powder diffractometer equipped with a CuKα (λα = 1.54178 Å) sealed tube (Bruker Corporation, Karlsruhe, Germany). UV–vis spectrum was performed on a UV-2450 spectrophotometer (Dongguan Hongcheng Optical Products Co., Dongguan, China). Emission spectra for the solid samples were determined with a Hitachi F-4500 fluorescence spectrophotometer (Hitachi, Tokyo, Japan).

### 3.2. Materials

The reagent Ni(OAc)_2_·4H_2_O was purchased from Alfa Aesar Co. (Ward Hill, MA, USA), and Cd(OAc)_2_·2H_2_O, Cu(OAc)_2_·H_2_O and Zn(OAc)_2_·2H_2_O from J. T. Baker Co. (Phillipsburg, NJ, USA). The ligand 4,4′-oxybis[N-(pyridin-3-ylmethyl)benzamide] (**L**) was prepared according to a published procedure.

### 3.3. Preparations

#### 3.3.1. {[Cd(**L**)(1,3-BDC)(H_2_O)]∙2H_2_O}_n_, **1**

A mixture of Cd(OAc)_2_·2H_2_O (0.027 g, 0.10 mmol), **L** (0.044 g, 0.10 mmol) and 1,3-H_2_BDC (0.017 g, 0.10 mmol) in 10 mL of EtOH/H_2_O (3:2) was sealed in a 23 mL Teflon-lined stainless steel autoclave, which was heated under autogenous pressure to 100 °C for two days, and then, the reaction system was cooled to room temperature at a rate of 2 °C per hour. The colorless crystals suitable for single-crystal X-ray diffraction were obtained. Yield: 0.055 g (72%). Anal. Calcd for C_34_H_32_N_4_O_10_ Cd (MW = 769.05): C, 53.09; H, 4.19; N, 7.28%. Found: C, 52.81; H, 4.18; N, 7.14%. FT-IR (cm^−1^): 3408(m), 3220(w), 3056(w), 2926(w), 1646(s), 1604(s), 1543(s), 1493(m), 1436(m), 1385(s), 1298(w), 1239(s), 1166(m), 1107(w), 1051(w), 859(m), 747(m), 601(w).

#### 3.3.2. {[Cd(**L**)(1,4-HBDC)(1,4-BDC)_0.5_]∙2H_2_O}_n_, **2**

Complex **2** was prepared using similar procedures to those for **1**, except that a mixture of Cd(OAc)_2_·2H_2_O (0.027 g, 0.10 mmol), **L** (0.044 g, 0.10 mmol) and 1,4-H_2_BDC (0.026 g, 0.15 mmol) in 10 mL of H_2_O was used. Colorless crystals. Yield: 0.035 g (42%). Anal. Calcd for C_38_H_33_N_4_O_11_Cd (MW = 834.10): C, 54.72; H, 3.99; N, 6.72%. Anal. Calcd. for **2**–H_2_O, C_38_H_31_N_4_O_10_Cd (MW = 816.09): C, 55.93; H, 3.83; N, 6.87%. Found: C, 55.57; H, 3.77; N, 6.91%. FT-IR (cm^−1^): 3523(m), 3299(m), 3060(w), 1693(s), 1636(m), 1560(s), 1499(s), 1395(s), 1282(m), 1250(s), 1166(w), 1110(w), 1052(w), 938(w), 846(m), 795(w), 748(m), 503(m).

#### 3.3.3. {[Cu_2_(**L**)_2_(1,3-BDC)_2_]∙1.5H_2_O}_n_, **3**

Complex **3** was prepared using similar procedures to those for **1**, except that a mixture of Cu(OAc)_2_·H_2_O (0.020 g, 0.10 mmol), **L** (0.044 g, 0.10 mmol) and 1,3-H_2_BDC) (0.017 g, 0.10 mmol) in 10 mL of H_2_O was used. Blue crystals were obtained. Yield: 0.015 g (11%). Anal. Calcd for C_68_H_55_Cu_2_N_8_O_15.5_ (MW = 1359.28): C, 60.09; H, 4.08; N, 8.24%. Found: C, 60.16; H, 3.80; N, 8.65%. FT-IR (cm^−1^): 3432(s), 2814(w), 2732(w), 2346(w), 2026(w), 1592(s), 1490(w), 1438(m), 1383(m), 1350(m), 1245(m), 1169(w), 1071(w), 978(w), 758(m), 703(m), 656 (m), 617(m), 539(m), 442(m).

#### 3.3.4. {[Ni(**L**)(1,3-BDC)(H_2_O)]∙2H_2_O}_n_, **4**

Complex **4** was prepared using similar procedures to those for **1**, except that a mixture of Ni(OAc)_2_·4H_2_O (0.025 g, 0.10 mmol), L (0.044 g, 0.10 mmol) and 1,3-H_2_BDC) (0.017 g, 0.10 mmol) in 10 mL of H_2_O was used. Green crystals were obtained. Yield: 0.059 g (82%). Anal. Calcd for C_34_H_32_N_4_O_10_Ni (MW = 715.33): C, 57.08; H, 4.51; N, 7.83%. Found: C, 57.04; H, 4.64; N, 7.75% FT-IR (cm^−1^): 3420(s), 3059(w), 1643(s), 1611(s), 1550(s), 1489(s), 1433(m), 1376(s), 1239(s), 1166(m), 1110(w), 1043(w), 987(w), 922(w), 861(m), 754(m), 728(m), 700(m).

#### 3.3.5. {[Zn(**L**)(1,3-BDC)]∙4H_2_O}_n_, **5**

Complex **5** was prepared using similar procedures to those for **1**, except that a mixture of Zn(OAc)_2_·2H_2_O (0.022 g, 0.10 mmol), **L** (0.044 g, 0.10 mmol), 1,3-H_2_BDC (0.017 g, 0.10 mmol) and NaOH (0.008 g, 0.10 mmol) in 10 mL of EtOH/H_2_O (1:4) was used. Colorless crystals were obtained. Yield: 0.043 g (58%). Anal. Calcd for C_34_H_34_N_4_O_11_Zn (MW = 740.04): C, 55.18; H, 4.63; N, 7.57%. Found: C, 54.92; H, 4.73; N, 7.35%. FT-IR (cm^−1^): 3468(m), 3316(m), 3242(m), 3065(m), 1617(s), 1545(m), 1494(s), 1435(m), 1384(m), 1330(s), 1239(s), 1170(m), 873(m), 847(m), 744(m), 655(m), 557(m).

#### 3.3.6. {[Zn(**L**)(1,4-BDC)]∙2H_2_O}_n_, **6**

Complex **6** was prepared using similar procedures to those for **1**, except that a mixture of Zn(OAc)_2_·2H_2_O (0.044 g, 0.20 mmol), **L** (0.044 g, 0.10 mmol) and 1,3-H_2_BDC (0.017 g, 0.10 mmol) in 10 mL of H_2_O was used. Colorless crystals were obtained. Yield: 0.038 g (53.98%). Anal. Calcd for C_34_H_30_N_4_O_9_Zn (MW = 704.01): C, 58.01; H, 4.30; N, 7.96%. Anal. Calcd. for **6**–H_2_O, C_34_H_28_N_4_O_8_Zn (MW = 686.01): C, 59.53; H, 4.11; N, 8.16%. Found: C, 59.91; H, 4.07; N, 8.23%. FT-IR (cm^−1^): 3614(w), 3530(w), 3252(m), 3068(m), 1641(s), 1599(s), 1490(s), 1347(s), 1231(s), 1174(m), 1055(m), 991(m), 874(m), 824(m), 752(m), 696(m), 501(m).

#### 3.3.7. [Cd_3_(**L**)_2_(1,4-BDC)_3_]_n_, **7**

Complex **7** was prepared using similar procedures to those for **2**, except that a mixture of Cd(OAc)_2_·2H_2_O (0.027 g, 0.10 mmol), **L** (0.044 g, 0.10 mmol) and 1,4-H_2_BDC (0.026 g, 0.15 mmol) in 10 mL of CH_3_OH was used. The reaction was performed at 80 °C. Colorless crystals were obtained. Yield: 0.010 g (6%). Anal. Calcd for C_76_H_56_N_8_O_18_Cd_3_ (MW = 1706.48): C, 53.49; H, 3.31; N, 6.57%. Calcd. for **7** + 2 H_2_O, C_76_H_64_N_8_O_22_Cd_3_ (MW = 1778.62): C, 52.38; H, 3.47; N, 6.43%. Found: C, 52.05; H, 3.18; N, 6.77%. FT-IR (cm^−1^): 3447(s), 2814(w), 2731(w), 2346(w), 2025(w), 1630(s), 1590(s), 1437(w), 1383(m), 1350(m), 1242(s), 1174(w), 1126(w), 1019(w), 756(m), 744(m), 618(m).

### 3.4. X-Ray Crystallography

Table 4 shows the crystal data of complexes **1**–**7**. The raw data of complexes **1**–**7** were collected by using a Bruker AXS SMART APEX II CCD diffractometer at 100 or 298 K, followed by data reduction, processed using standard methods [25]. Some of the heavier atoms were found by manipulating the direct or Patterson methods, and the remaining atoms were located in a series of alternating difference Fourier maps and least-square refinements. Moreover, the hydrogen atoms except those of the water molecules were added by using the HADD command in SHELXTL 6.1012 [26].

## 4. Conclusions

Seven non-entangled 1D and 2D CPs containing the angular **L** and the isomeric dicarboxylate ligands have been synthesized successfully under hydrothermal reactions. Complexes **1** and **4** adopt the same topology of (6^4^·8·10)(6)-2,4L3, and **2** and **3** show the topologies of (4^2^·8^2^·10^2^)(4^2^·8^4^)_2_(4)_2_ and (4·5·6)(4·5^5^·6^3^·7)-3,5L66, whereas **5**–**7** exhibit a (8^4^·12^2^)(8)_2_-2,4L2 topology with interdigitation, a (4^3^·6^2^·8)(4)-2,4C3 topology and a (3^6^·4^6^·5^3^)-**hxl** topology, respectively. The combination of the angular **L** with the dicarboxylic acid and metal salt is thus not likely to form the entangled CPs. Structural comparisons indicate that the identities of metal ions and dicarboxylate ligands play important roles in determining the structural diversity. Complexes **1** and **5** are two 2D CPs with the same organic ligands but different metal ions, resulting in different topological structures. The metal–ligand distances, coordination geometries, packing interactions and topological structures are given in Table 5. The results reveal that the quenching effect of complex **1** by the Fe^3+^ ions can be ascribed to competitive absorption and that of the interdigitated **5** is due to framework collapse. This investigation oversees the metal effect on the structural stabilities of **L** and 1,3-BDC^2−^-based Cd(II), **1**, and Zn(II), **5**, CPs that attack Fe^3+^ ions.

## Figures and Tables

**Figure 1 molecules-30-03283-f001:**
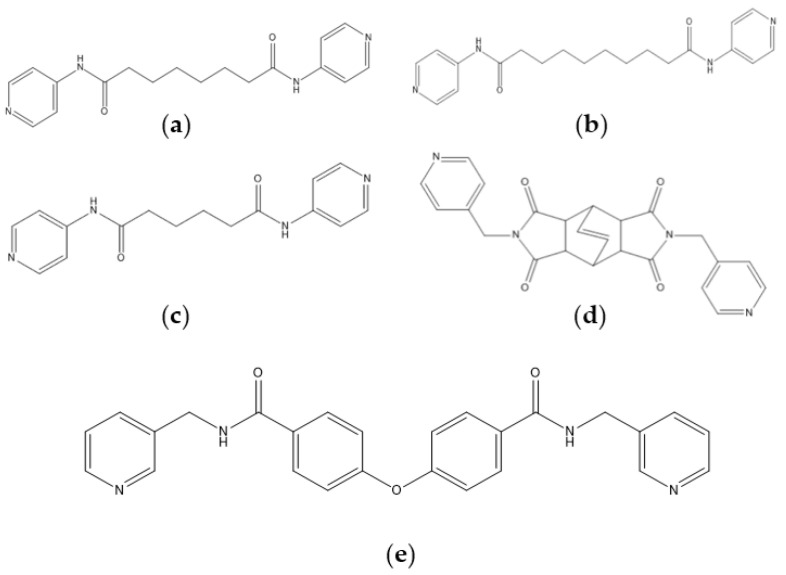
Structures of (**a**) *N,N′*-di(4-pyridyl)suberoamide, (**b**) *N,N′*-di(4-pyridyl)sebacoamide, (**c**) *N,N′*-di(4-pyridyl)adipoamide, (**d**) *N,N′*-bis(4-pyridylmethyl)bicyclo(2,2,2,)oct-7-ene-2,3,5,6-tetracarboxylic diamide and (**e**) **L**.

**Figure 2 molecules-30-03283-f002:**
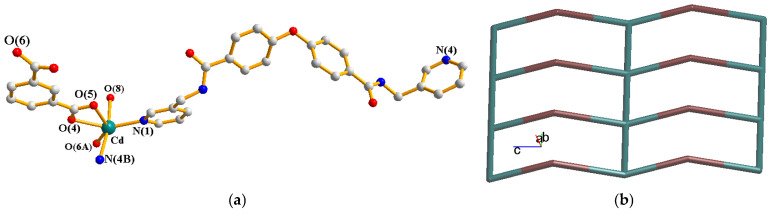
(**a**) Coordination environment of Cd(II) ion in **1**. Symmetry transformations used to generate equivalent atoms: (A) x, y + 1, z; (B) x + 1, y − 1, z + 1. (**b**) A drawing showing the 2D net with the 2,4L3 topology.

**Figure 3 molecules-30-03283-f003:**
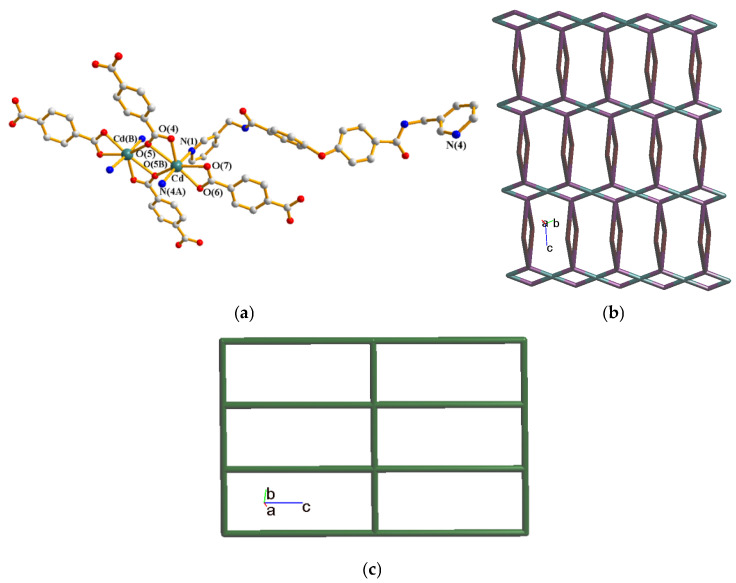
(**a**) Coordination environments of the Cd(II) ions in **2**. Symmetry transformations used to generate equivalent atoms: (A) −x + 1, −y + 1, −z + 1; (B) −x + 1, −y + 1, −z + 2. (**b**) A drawing showing the 2D net with the (4^2^·8^2^·10^2^)(4^2^·8^4^)_2_(4)_2_ topology. (**c**) A drawing showing the 2D net with the **sql** topology.

**Figure 4 molecules-30-03283-f004:**
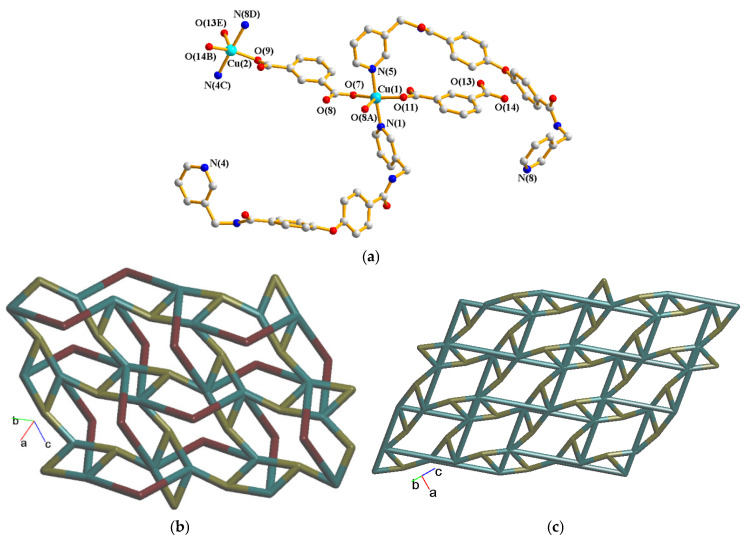
(**a**) Coordination environment of Cd(II) ion in **3**. Symmetry transformations used to generate equivalent atoms: (A) −x, −y + 1, −z; (B) x, y − 1, z + 1; (C) x + 1, y, z; (D) x + 1, y − 1, z + 1; (E) − x + 1, −y + 1, −z. (**b**) A drawing showing the 2D net with the (4·6·8)(4·6^5^·8^4^)(6) topology. (**c**) A drawing showing the 2D net with the 3,5L66 topology.

**Figure 5 molecules-30-03283-f005:**
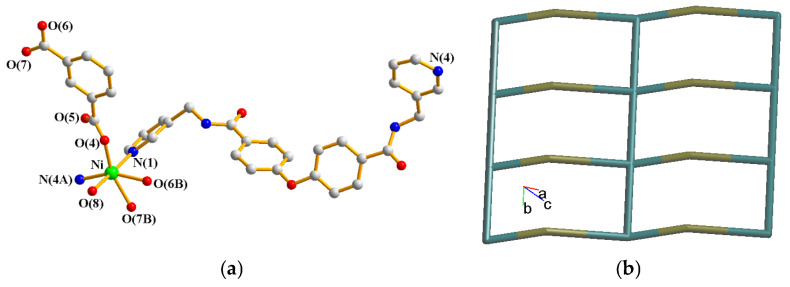
(**a**) Coordination environment of Ni(II) ion in **4**. Symmetry transformations used to generate equivalent atoms: (A) x + 1, y − 1, z + 1; (B) x, y + 1, z. (**b**) A drawing showing the 2D net with the 2,4L3 topology.

**Figure 6 molecules-30-03283-f006:**
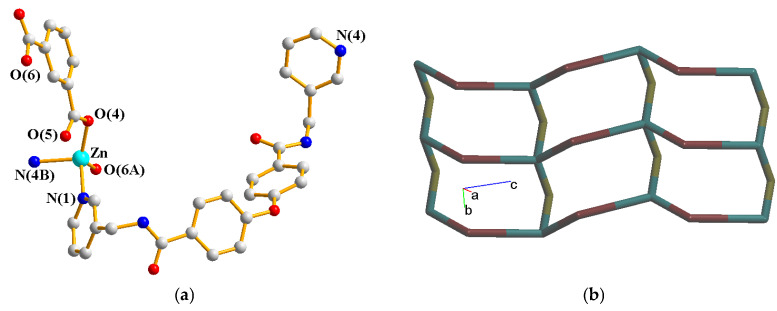
(**a**) Coordination environment of Zn(II) ion in **5**. Symmetry transformations used to generate equivalent atoms: (A) x, y + 1, z; (B) x − 1/2, −y + 7/2, z − 1/2. (**b**) A drawing showing the 2D net with the 2,4L2 topology. (**c**) A drawing showing a pair of interdigitated 2D layers. (**d**) Another view of the interdigitated 2D layers.

**Figure 7 molecules-30-03283-f007:**
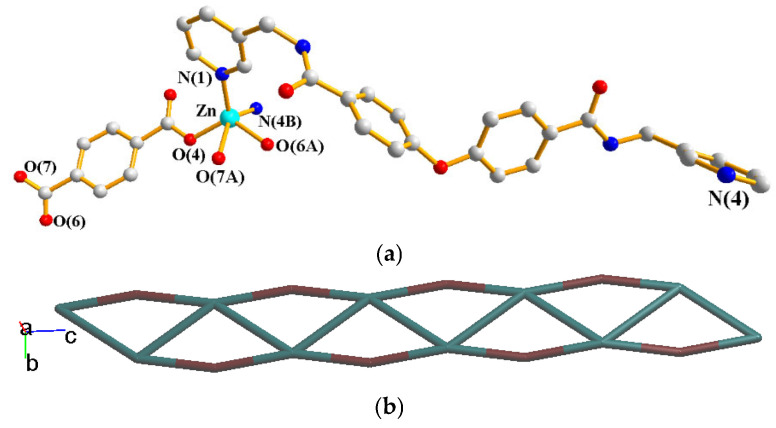
(**a**) Coordination environment of Zn(II) ion in **6**. Symmetry transformations used to generate equivalent atoms: (A) x, −y + 3/2, z + 1/2; (B) x, y, z − 1. (**b**) A drawing showing the 1D chain with the 2,4C3 topology.

**Figure 8 molecules-30-03283-f008:**
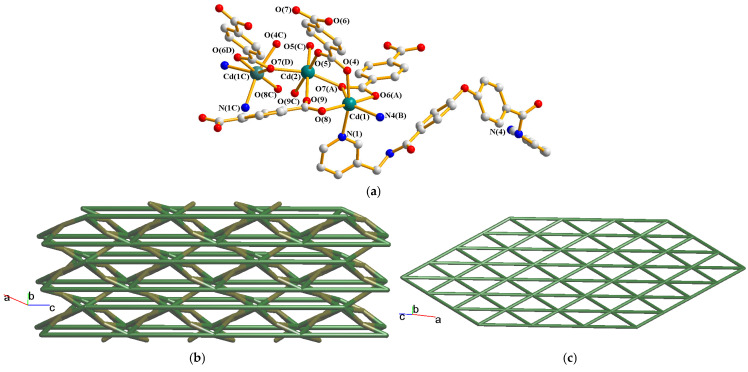
(**a**) Coordination environments of the Cd(II) ions in **7**. Symmetry transformations used to generate equivalent atoms: (A) x, −y, z + 1/2; (B) x, y, z − 1; (C) −x, y, −z + 1/2. (D) −x, −y, −z. (**b**) A drawing showing the 2D net with the(4^2^·6·7^3^)(4^3^·5^2^·6^4^·7)_2_(4^3^·5^3^)_2_(4^8^·6^6^·7) topology. (**c**) A drawing showing the 2D net with the **hxl** topology.

**Figure 9 molecules-30-03283-f009:**
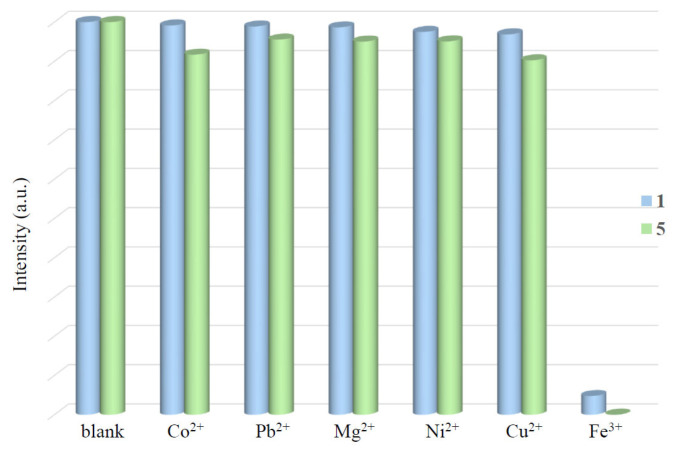
The average intensities of complexes 1 and 5 upon immersion in Fe^3+^ solution.

**Figure 10 molecules-30-03283-f010:**
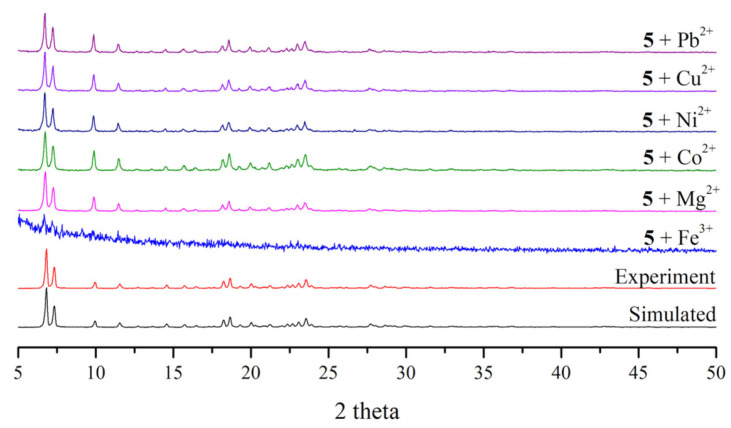
Luminescence intensities of complex **5** upon immersion in the aqueous solutions of different metal ions.

**Figure 11 molecules-30-03283-f011:**
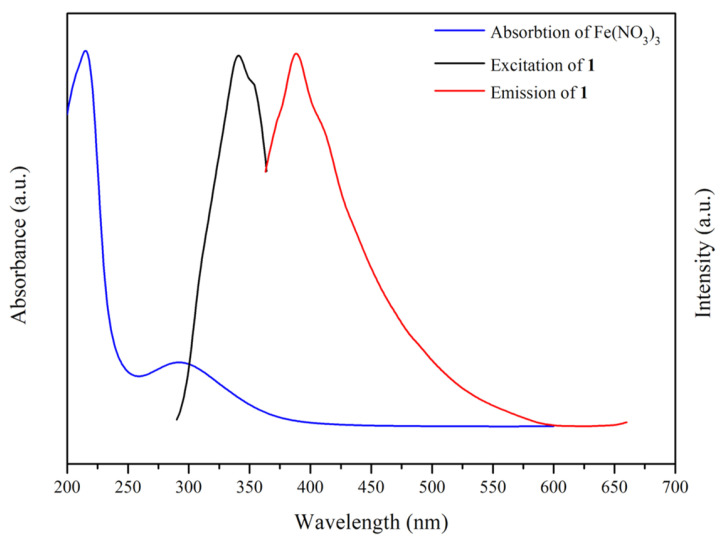
UV–vis absorption spectra of Fe^3+^ (blue line) along with the excitation and emission spectrum of 1 (black and red line).

**Figure 12 molecules-30-03283-f012:**
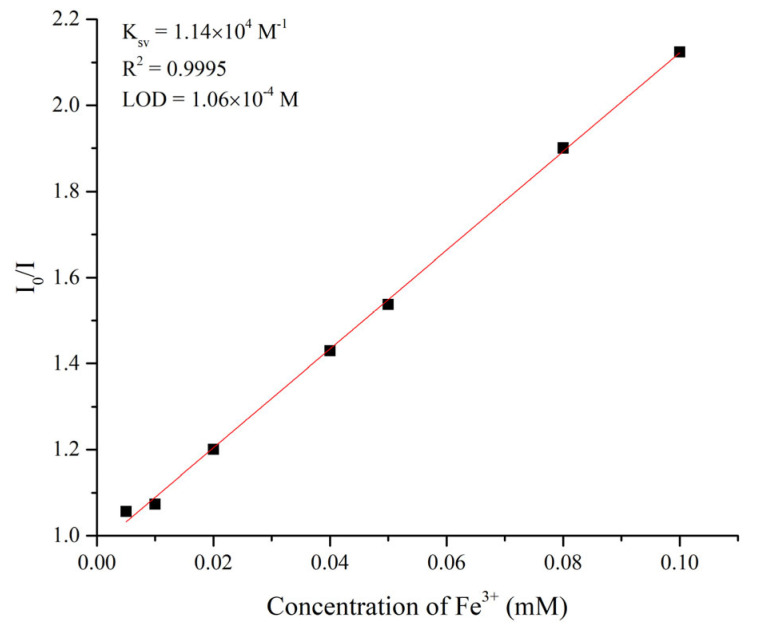
The SV plot of complex **1**.

**Table 1 molecules-30-03283-t001:** Ligand conformations and bonding modes of complexes **1**–**7**.

	Ligand Conformation	Coordination Mode
**1**	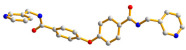 *syn–anti*	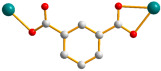 μ_2_-κ^2^O,O′;κO″
**2**	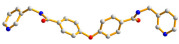 *syn–syn*	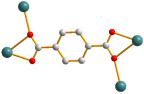 μ_4-_κ^2^O,O′;κO′;κ^2^O″,O‴; κO‴ 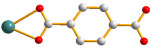 κ^2^O,O′
**3**	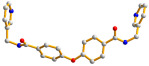 *anti–anti* 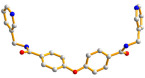 *syn–anti*	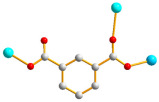 μ_3_-κO;κO′;κO″
**4**	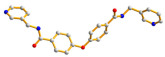 *syn–anti*	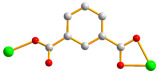 μ_2_-κ^2^O,O′;κO″
**5**	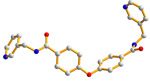 *anti–anti*	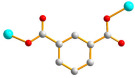 μ_2_-κO;κO′
**6**	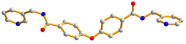 *anti–anti*	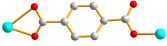 μ_2_-κ^2^O,O′;κO″
**7**	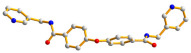 *syn–anti*	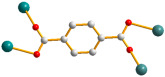 μ_4_-κO;κO′;κO″;κO‴ 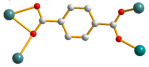 μ_4-_κO;κO′;κ^2^O″,O‴; κO‴

**Table 2 molecules-30-03283-t002:** Thermal decompositions of complexes **1**–**7**.

Complex	Weight Loss of Solvent °C (Calc/Found),%	Weight Loss of Ligand °C (Calc/Found),%
**1**	3 H_2_O 90–130 (7.38/7.02)	**L** + 1,3-BDC^2−^ 325–800 (82.60/78.28)
**2**	1 H_2_O ~220 (2.65/2.16)	**L** + 1,4-BDC^2−^ 275–800 (82.60/82.50)
**3**	1.5 H_2_O ~230 (1.99/3.31)	**L** + 1,3-BDC^2−^ 245–800 (88.66/89.94)
**4**	3 H_2_O 80–180 (7.60/7.55)	**L** + 1,3-BDC^2−^ 335–800 (82.28/84.44)
**5**	4 H_2_O ~170 (9.11/9.72)	**L** + 1,3-BDC^2−^ 310–795 (72.13/81.35)
**6**	2 H_2_O ~335 (4.87/5.11)	**L** + 1,4-BDC^2−^ 345–800 (84.90/85.80)
**7**		**L** + 1,4-BDC^2−^ 345–800 (80.24/76.99)

**Table 3 molecules-30-03283-t003:** Luminescent properties of **L**, 1,3-H_2_BDC, 1,4-H_2_BDC, **1**, **2**, **5**, **6** and **7**.

Compound	Excitation λ_ex_ (nm)	Emission λ_em_ (nm)	Complex	Excitation λ_ex_ (nm)	Emission λ_em_ (nm)
1,3-H_2_BDC	280,310	355	**1**	340	388
1,4-H_2_BDC	280,330	385	**2**	330	385
**L**	340	398	**5**	341	422
			**6**	335	435
			**7**	321	421

**Table 4 molecules-30-03283-t004:** Crystal data for complexes **1**–**7**.

Complex	1	2		3		4
Formula	C_34_H_32_CdN_4_O_10_	C_38_H_33_CdN_4_O_11_		C_68_H_55_Cu_2_N_8_O_15.5_		C_34_H_32_NiN_4_O_10_
Formula weight	769.03	834.08		1359.28		715.34
Crystal system	Triclinic	Triclinic		Triclinic		Triclinic
Space group	*P*ī	*P*ī		*P*ī		*P*ī
a, Å	9.3402(5)	9.2561(5)		11.8074(2)		9.3094(2)
b, Å	10.1891(6)	9.9219(5)		16.1994(3)		10.0075(1)
c, Å	18.4381(11)	22.6929(11)		16.9383(3)		18.1094(3)
α, °	81.322(3)	82.3318(13)		73.9272(12)		79.3537(8)
*β*, °	77.049(3)	80.9252(15)		78.3137(12)		78.0679(7)
γ, °	70.622(3)	62.7198(13)		78.6938(12)		73.4443(7)
V, Å^3^	1607.61(16)	1824.65(16)		3014.74(10)		1567.95(5)
Z	2	2		2		2
D_calc_, Mg/m^3^	1.589	1.518		1.497		1.515
F(000)	784	850		1402		744
µ (Mo K_α_), mm^−1^	0.746	0.666		0.785		0.687
Range (2θ) for data collection, deg	2.126 to 28.499	2.315 to 25.091		1.590 to 28.294		2.142 to 28.331
Independent reflection	8049[R(int) = 0.0210]	6465[R(int) = 0.0725]		14,951[R(int) = 0.0380]		7808[R(int) = 0.0216]
Data/restraint/parameter	8049/0/450	6465/0/649		14,951/0/842		7808/0/442
quality-of-fit indicator ^c^	1.035	1.057		1.023		1.025
Final R indices [I > 2σ(I)] ^a,b^	R1 = 0.0259,wR2 = 0.0629	R1 = 0.0531,wR2 = 0.1280		R1 = 0.0462,wR2 = 0.1016		R1 = 0.0301,wR2 = 0.0733
R indices (all data)	R1 = 0.0283,wR2 = 0.0645	R1 = 0.0718,wR2 = 0.1426		R1 = 0.0840,wR2 = 0.1161		R1 = 0.0377,wR2 = 0.0770
**Complex**	**5**		**6**		**7**	
Formula	C_34_H_34_ZnN_4_O_11_		C_34_H_30_ZnN_4_O_9_		C_76_H_56_Cd_3_N_8_O_18_	
Formula weight	740.02		703.99		1706.48	
Crystal system	Monoclinic		Monoclinic		Monoclinic	
Space group	*P*2_1_*/n*		*P*2_1_/*c*		*C*2/*c*	
a, Å	13.2504(4)		14.6873(2)		41.100(2)	
b, Å	9.5834(3)		12.5880(2)		9.7849(4)	
c, Å	26.5625(9)		17.7902(3)		18.1394(7)	
α, °	90		90		90	
*β*, °	95.3579(6)		97.0769(10)		113.9416(11)	
γ, °	90		90		90	
V, Å^3^	3358.27(19)		3264.06(9)		6667.3(5)	
Z	4		4		4	
D_calc_, Mg/m^3^	1.464		1.433		1.700	
F(000)	1536		1456		3424	
µ (Mo K_α_), mm^−1^	0.799		0.814		1.031	
Range (2θ) for data collection, deg	1.540 to 28.393		1.397 to 25.999		2.151 to 26.051	
Independent reflection	8416[R(int) = 0.0445]		6400[R(int) = 0.0572]		5902[R(int) = 0.0715]	
Data/restraint/parameter	8416/0/451		6400/0/442		5902/0/474	
quality-of-fit indicator ^c^	1.036		1.056		1.076	
Final R indices [I > 2σ(I)] ^a,b^	R1 = 0.0424,wR2 = 0.1018		R1 = 0.0507,wR2 = 0.0977		R1 = 0.0378,wR2 = 0.0730	
R indices (all data)	R1 = 0.0721,wR2 = 0.1134		R1 = 0.1236,wR2 = 0.1201		R1 = 0.0737,wR2 = 0.0922	

^a^ R_1_ = Σ||F_o_| − |F_c_||/Σ|F_o_|. ^b^ wR_2_ = [Σw(F_o_^2^ − F_c_^2^)^2^/Σw(F_o_^2^)^2^]^1/2^. w = 1/[σ^2^(F_o_^2^) + (ap)^2^ + (bp)], *p* = [max(F_o_^2^ or 0) + 2(F_c_^2^)]/3. a = 0.0287, b = 1.3165 for **1**; a = 0.0757, b = 3.3200 for **2**; a = 0.0489, b = 1.4318 for **3**; a = 0.0367, b = 0.555 for **4**; a = 0.0514, b = 0.5942 for **5**; a = 0.0460, b = 0.0000 for **6**; a = 0.0229, b = 41.0443 for **7**. ^c^ quality-of-fit = [Σw(|F_o_^2^| − |F_c_^2^|)^2^/(N_observed_ − N_parameters_)]^1/2^.

**Table 5 molecules-30-03283-t005:** Metal–ligand distances, coordination geometries, packing interactions and topologies of complexes **1**–**7**.

Complex	Metal–Ligand Distance	Coordination Geometry and Packing Interactions	Topology
**1**	Cd-N(1) = 2.2697(14) Cd-O(6A) = 2.2733(13) Cd-O(4) =2.3338(13) Cd-O(8) = 2.3393(14) Cd-N(4B) = 2.3427(15) Cd-O(5) = 2.4849(12)	Distorted octahedral N-H---O and O-H---O	2D net with (6^4^·8·10)(6)-2,4L3
**2**	Cd-N(4A) = 2.277(5) Cd-N(1) = 2.286(6) Cd-O(4) = 2.297(4) Cd-O(5B) = 2.337(4) Cd-O(6) = 2.355(3) Cd-O(7) = 2.434(3) Cd-O(5) = 2.604(4)	Distorted pentagonal bipyramidal N-H---O and O-H---O	2D net with (4^2^·8^2^·10^2^)(4^2^·8^4^)_2_(4)_2_
**3**	Cu(1)-O(7) = 1.9504(17) Cu(1)-O(11) = 1.9616(16) Cu(1)-N(5) = 2.016(2) Cu(1)-N(1) = 2.023(2) Cu(1)-O(8A) = 2.3684(19) Cu(2)-O(14B) = 1.9368(16) Cu(2)-O(9) = 1.9755(15) Cu(2)-N(4C) = 2.011(2) Cu(2)-N(8D) = 2.026(2) Cu(2)-O(13E) = 2.396(2)	Distorted square pyramidal N-H---O and O-H---O	2D net with (4·5·6)(4·5^5^·6^3^·7)-3,5L66
**4**	Ni-O(4) = 2.0344(10) Ni-N(4A) = 2.0658(12) Ni-N(1) = 2.0895(13) Ni-O(8) = 2.0896(11) Ni-O(7B) = 2.1230(10) Ni-O(6B) = 2.1563(10)	Distorted octahedral N-H---O and O-H---O	2D net with (6^4^·8·10)(6)-2,4L3
**5**	Zn-O(4) = 1.9293(16) Zn-O(6A) = 1.9419(14) Zn-N(1) = 2.043(2) Zn-N(4B) = 2.111(2)	Distorted tetrahedral N-H---O and O-H---O	2D net with (8^4^·12^2^)(8)_2_-2,4L2
**6**	Zn-O(4) = 1.939(2) Zn-N(1) = 2.033(3) Zn-O(6A) = 2.044(4) Zn-N(4B) = 2.063(3) Zn-O(7A) = 2.421(4)	Distorted square pyramidal N-H---O and O-H---O	1D net with (4^3^·6^2^·8)(4)-2,4C3
**7**	Cd(1)-O(4) = 2.247(4) Cd(1)-O(6A) = 2.341(3) Cd(1)-O(8) = 2.374(3) Cd(1)-N(4B) = 2.378(4) Cd(1)-N(1) = 2.382(4) Cd(1)-O(7A) = 2.464(3) Cd(2)-O(5) = 2.166(3) Cd(2)-O(5C) = 2.167(3) Cd(2)-O(9C) = 2.272(3) Cd(2)-O(9) = 2.273(3) Cd(2)-O(7A) = 2.402(3) Cd(2)-O(7D) = 2.402(3)	Distorted octahedral N-H---O	2D net with (3^6^·4^6^·5^3^)-**hxl**

## Data Availability

Data are contained within this article or its Appendix A.

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
