# Peer review of "Coordination Polymers Bearing Angular 4,4′-Oxybis[*N*-(pyridin-3-ylmethyl)benzamide] and Isomeric Dicarboxylate Ligands: Synthesis, Structures and Properties"

_molecules, 2025, doi:10.3390/molecules30153283_

Round 1

Reviewer 1 Report

Comments and Suggestions for Authors

The manuscript by Huang and co-authors reports the synthesis of several non-entangled 1D and 2D coordination polymers using angular ligands and dicarboxylate linkers. The resulting structures show a variety of topologies, influenced by both the metal ions and the ligands used. Changes in metal centers lead to differences in structural stability and response to external conditions. While the study focuses on basic investigations into how metal ions affect structure and properties, the methods are appropriate, the data convincing, and the findings relevant to the readers of Molecules. That said, a few minor issues should be addressed as follows:

1. A tabulated comparison of metal–ligand distances, coordination geometries, and packing interactions across complexes would better support conclusions about metal-dependent structural diversity.

2. Complex 1 shows high Fe3+ quenching efficiency. Given the partial overlap in UV-vis absorption, how do the authors distinguish between static and dynamic quenching mechanisms? Was lifetime measurement or temperature dependence tested?

3. The manuscript attributes the observed luminescence of complexes 1, 2, 5, 6, and 7 primarily to intraligand (IL) or ligand-to-ligand charge transfer (LLCT) transitions, based on the d¹⁰ configuration of Zn(II) and Cd(II) centers. can the authors discuss more on how can LMCT contributions or metal-perturbed IL states be confidently ruled out, especially given the diverse coordination geometries?  

4. Lastly, it would be helpful for the authors to include the extended packing structures of each coordination polymer. While the simplified geometrical representations generated by ToposPro are useful for topological analysis, visualizing the full molecular stacking would provide a clearer understanding of the overall packing and spatial arrangement.

Reviewer 2 Report

Comments and Suggestions for Authors

In this manuscript, the authors presented the synthesis, structural characterization, and various properties of seven different coordination polymers. While the work shows potential for publication in this journal, it requires substantial improvements before it can be considered for acceptance. I encourage the authors to carefully address the following points and make the necessary edits and additions.

  1. The authors should consider citing relevant work by other researchers in the field of MOFs. In its current form, the manuscript contains overall 35% self-citations, and more than 50% self-citations within the MOF-related references, which undermines the breadth and objectivity of the literature coverage.
  2. In its current form, the introduction presents the background of the work, primarily based on two previously published studies by the same author group. The authors should expand the background discussion to include contributions from other researchers as well, since many groups are actively working in this area. 
  3. Including a schematic representation of the angular ligands used in the discussion would greatly enhance the clarity of the background section.
  4. The Introduction section lacks a clear motivation for the work. It should elaborate on why entangled and non-entangled networks are of interest, what advantages they offer, what alternative strategies exist for designing such frameworks, and the potential applications of these materials. 
  5. Since the major contribution of this work to the MOF field lies in the structural diversity arising from different ligands, a more comprehensive structural discussion is essential. Specifically, a clear description of how each ligand extends the framework in particular directions, accompanied by appropriate schematic or crystallographic figures, would significantly enhance the clarity and impact of the manuscript.
  6. It is not very clear what the authors intended to convey in the following phrases:
  • Page 2, line 71: "while the 1,3-BDC 2- ligands as linkers"
  • Page 2, line 86: "ligands serve as linkers"
  • Page 3, line 88: "while the organic ligands as linkers,"
  • Page 4, line 109: "ligands serve as linkers," And a few more similar phrases.
  1. Authors should include details about how the interdigitated 2D layered structure is formed for 5.
  2. For luminescence studies, the authors have used a pair of excitation wavelengths without providing a clear rationale or method for selecting them. A justification for the choice of excitation wavelengths is necessary to validate the experimental design.
  3. The authors have performed Fe(III) ion detection studies only for CP1 and CP5, without providing a rationale for excluding the other coordination polymers. A clear and logical explanation for this selective investigation is necessary.
  4. The crystallinity of CP1 exhibits significant degradation, and for CP5, it is completely lost after soaking in the Fe(III) solution, as evident from Figures S29–S31. This indicates substantial structural breakdown, which could be a major contributor to the observed luminescence quenching. Although the authors claim there is no change in the PXRD pattern of CP1, a closer examination of the PXRD patterns reveals signs of structural degradation.
  5. Although the authors attribute luminescence quenching to competitive energy absorption between CP1 and Fe(III) ions, the evidence provided is still insufficient to firmly establish this mechanism. Stern-Volmer analysis and the minor overlap in absorption spectra are not sufficient. Additional supporting data, such as luminescence lifetime measurements or control experiments with similar ions, would strengthen the mechanistic understanding and support the authors' interpretation more convincingly.
  6. "nides" at page 4, line 107. Should be "nodes."
  7. It is not very suitable to mention the 12-hour soaking experiment as "metal sensing."
  8. Right now, the manuscript gives the impression that all these coordination polymers were synthesized from a somewhat arbitrary selection of metal ions and dicarboxylic ligands, simply combined with an angular ligand. The subsequent characterization of these CPs appears to be a routine process. To be considered for publication, this perception needs to change.
  9. The manuscript should be restructured to demonstrate that every step, from the choice of precursors to the characterization experiments, is driven by sound scientific reasoning and logical intuition. This means clearly articulating the rationale behind the selection of specific metal ions and ligands, highlighting why this particular combination was expected to yield the observed coordination polymers. Furthermore, the characterization experiments shouldn't just be presented as routine; their selection should be justified by the specific questions they aim to answer regarding the formed CPs.

Round 2

Reviewer 2 Report

Comments and Suggestions for Authors

Authors have done a noteworthy job in reshaping and updating their manuscript. But a few more points still need to be addressed before the manuscript can be accepted for publication. I encourage the authors to go through the following points and make the necessary edits and additions.

  1. The addition made on page 1, lines 35-39, is confusing. At first, sentence entanglement is described as fascinating, and in the second sentence, it becomes a limitation. Overall, the flow is lost. Please reframe the portion.
  2. Authors may consider including images for N,N’-di(4-pyridyl)suberoamide, N,N'-di(4-pyridyl)sebacoamide, N,N'-di(4-pyridyl)adipoamid, N,N'-bis(4-pyridyl- methyl)bicyclo(2,2,2,)oct-7-ene-2,3,5,6-tetracarboxylic diamide etc which they have used in the discussion. With a figure, the section will be clearer to readers.
  3. Authors may consider citing https://doi.org/10.1021/acs.cgd.7b01270 where ligand rigidity influences the formation of an interpenetrated framework and flexibility influences the formation of a “non-interpenetrated” framework.
  4. The work still lacks a clear indication of motivation.
  5. The number of CPs made from 1,3-BDC and 1,4-BDC are different. What is the thought of authors about not having a Ni(II) or Cu(II) CPs from 1,4-BDC?
  6. Only a topological image is not sufficient to describe all the structure, especially 3D structures. Authors should provide 1D/2D figures of metal-individual ligand structure, like (metal-L)n structures, for all the CPs.
  7. In the current form of the manuscript authors included a discussion of hydrogen bonding between guest water molecules and frameworks. One representative figure of such bonding for individual frameworks will enhance the understanding.
  8. “…show emissions at 398, 355 and 385 nm upon excitations at 340, 280, 310 and 330 nm, respectively, …” in this part, it looks like 4 excitations give three emissions and ends with ‘respectively’. It is not possible to understand which excitation wavelength gives which emission.
  9. Blue and red shifts of emission wavelengths are not explained properly. I suggest authors to include the logical reasons for shifts with proper citations.
  10. The authors mentioned that luminescence quenching occurred due to competitive energy absorption between Fe3+ and CP1. If this logic is true, then why is 12 hours needed for the quenching to manifest? This suggests that the quenching is not primarily due to competitive energy absorption; rather, CP1 is likely undergoing partial breakdown during the 12-hour soaking in Fe3+ solution, a hypothesis supported by the PXRD pattern.
  11. Instrumental details for emission and UV-vis spectroscopy are required.
  12. Also, I like to encourage authors to include details about experimental details of excitation and emission spectroscopy.
